# Cooperative behaviors and social interactions in the carnivorous bat *Vampyrum spectrum*

**Marisa Tietge**[1,2]*, **Eduardo Artavia Durán**[3], **Mirjam Knörnschild**[1,2,4]

**1** Museum für Naturkunde - Leibniz Institute for Evolution and Biodiversity Science, Berlin, Germany, **2** Humboldt-Universität zu Berlin, Institute for Biology, Berlin, Germany, **3** Manzú Conservación de Murciélagos, Liberia, Guanacaste, Costa Rica, **4** Smithsonian Tropical Research Institute, Ancon, Panama

* marisa.tietge@gmail.com

## Abstract

Bats exhibit a diverse array of social behaviors, yet detailed studies on the intricacies of these interactions, particularly in rare species like the spectral bat (*Vampyrum spectrum*), remain scarce. This study presents the first comprehensive description of prey provision and other social behaviors in a wild social group of *V. spectrum*. Over several months, we conducted extensive video recordings in a hollow tree, used as a day- and night-roost, to document these behaviors, aiming to elucidate key social interactions among the bats and their role within their ecological niche. We observed various remarkable social behaviors, including food provision between family members. Our findings support the hypothesis that prey provision as a form of biparental care may serve as a method for adults to transition young bats from milk to a carnivorous diet, ensuring adequate food intake and allowing them to practice how to handle large prey items. We also challenge the notion that *V. spectrum* forages exclusively solitarily, as we documented several instances of synchronized roost departures and returns and thus presumably cooperative foraging. This indicates a more complex social structure and behavioral ecology than previously understood. Our comprehensive analysis of observational data enhances our understanding of the social dynamics of *V. spectrum*, providing new insights into the evolution of cooperative behaviors in bats.

## Introduction

Cooperative behaviors are observed across many animal species and play a crucial role in their survival and social structures [1–3]. Such behaviors often involve individuals working together to achieve shared goals, like defending territories, caring for young, or securing resources [3–5]. These interactions are especially well-documented in social animals, where cooperation can range from kinship-based behaviors to complex alliances among unrelated individuals [6–8]. In addition to kinship, alliances among non-relatives are widespread; for instance, in species like

**Data availability statement:** All relevant data are within the paper and its Supporting Information file.

**Funding:** This work was supported by a grant from the Leibniz Foundation (P122/2020) to Mirjam Knörnschild.

**Competing interests:** The authors have declared that no competing interests exist.

baboons and dolphins, unrelated individuals cooperate to hunt more effectively, which enhances foraging success [9–12]. This behavior allows access to larger or more challenging prey than would be possible alone, illustrating how collaborative behaviors can enhance survival and fitness within an ecological niche [13].

Bats, with over 1,480 species [14], exhibit cooperative behaviors on many levels. As one of the most ecologically significant and diverse orders of mammals, bats occupy a broad range of habitats and exhibit various dietary preferences – including insectivory, frugivory, nectarivory, piscivory, and even carnivory [15,16]. Their social structures are equally varied, spanning solitary roosting, complex communal arrangements, such as harems, social monogamy, and promiscuity [17,18], making them an exceptional case for studying the dynamics of cooperation across different social systems. Most bat species are highly gregarious, presumably due to ecological constraints like roost limitation, physiological demands like social thermoregulation, and their longevity [18]. Bats exhibit complex communication, group-level recognition, and various other social interactions among individuals [19]. Some remarkable social behaviors have evolved, like roost sharing and defense in tent-making bats where individuals construct tents from large leaves. These shelters provide protection from predators and harsh weather. Bats within these groups cooperate in the maintenance and defense of their roosts [20–22]. Previous studies have shown that bats from various social systems can show cooperative behavior when it comes to the availability of food and prey capture using auditory, visual, and olfactory cues, especially when living in fission-fusion systems [23–29]. Behavioral evidence also suggests that bats use calls to distinguish between familiar and unfamiliar conspecifics, like the Spix's disk-winged bats (*Thyroptera tricolor*) that uses social calls to stay in touch with group members while foraging [30]. This behavior is similar to "grooming at a distance" in primates, where individuals that frequently groom each other maintain vocal contact when separated during foraging (reviewed in [31]. Maintaining contact while foraging may help bats ensure they later roost with preferred group members and reinforce cooperative behavior [30].

Understanding the social behaviors of bats is crucial for several reasons. Social interactions can impact reproductive success, foraging efficiency, predator avoidance, and disease transmission [32–40]. These behaviors are also integral to the conservation and management of bat populations, many of which are threatened by habitat loss, climate change, and human activities [41]. Bat species developed a lot of different social behaviors, most of the them are still unknown because bats are often hard to observe due to their nocturnal and highly mobile lifestyle. Especially bat species that are rare to find and/or are critically endangered are understudied.

The spectral bat (*Vampyrum spectrum*) is widely recognized as the largest carnivorous bat species in the New World, with individuals weighing approximately 180 grams and boasting a wingspan of up to 900 millimeters [42]. This rare species can be found in lowland tropical dry forests, evergreen forests, and occasionally in cloud forests, deciduous forests, or swampy areas [43]. *Vampyrum spectrum* is considered near threatened by the International Union for Conservation of Nature (IUCN) and is

listed as a species of special concern or endangered in several countries throughout its range with a decreasing population trend [44]. Belonging to the family of leaf-nosed bats (Phyllostomidae), these formidable predators have a diverse diet that includes a variety of birds, rodents, and small mammals, including other bat species [42,45]. *Vampyrum spectrum* typically roosts in small groups within hollow trees or caves. These groups exhibit a socially monogamous structure, usually comprising a single male, a single female, and their recent offspring who have not yet dispersed [43]. This socially monogamous social structure is relatively rare among mammals, making this species a particularly interesting subject for study.

The primary goals of this study were to get insights into the social behaviors of *Vampyrum spectrum* within the context of their ecological niche as carnivorous bats. By conducting extensive video recordings in their roost, we aimed to document and analyze key behaviors, with a focus on social interactions and cooperative behaviors. This research not only aims to shed light on the social behaviors of *V. spectrum* but also contributes to the broader understanding of social behavior in bats.

## Methods

The study was approved and the field work permit was granted by el Ministerio del Ambiente y Energía, Sistema Nacional de Áreas de Conservación–Área de Conservación Guanacaste Sitio Patrimonio Natural de la Humanidad; Permit no. R-SINAC-ACG-PI-057–2024.

### Study site

The study was conducted in the tropical dry forest of "La Estación Experimental Forestal Horizontes" (N 10° 42' 47.883973, E −85° 35' 43.759147) in Guanacaste, Costa Rica. The focal roost is located within a partially hollow yet living tree of the species *Manilkara chicle*, amidst typical dry forest vegetation of Guanacaste, Costa Rica and adjacent to a dried riverbed. This area of tropical dry forest is characterized by a stratified vegetation structure comprising canopy, understory, shrub, and ground layers, each supporting distinct plant and animal communities. The canopy layer ranges from 20 to 30 meters in height and is composed of deciduous trees with broad crowns. The understory consists of trees reaching 10–20 meters, typically with light canopies and slender trunks. Beneath this, the shrub layer (2–5 m) is dominated by thorny, multi-stemmed plants. The roost site is located within a mature tree reaching a height of approximately 20 meters, with a maximum diameter at breast height (DBH) of 1m. The tree forms part of a structurally interconnected canopy, although it is not the tallest individual in the immediate vicinity. The roost itself is situated within a hollow trunk, the entrance of which is spanned approximately 2 meters above ground level. This aperture is roughly 80 cm wide at its base, progressively narrowing to approximately 50 cm with height, and lacks any secondary openings along the trunk.

This roost was first identified by M.T. in December 2022, housing four *V. spectrum* individuals (Fig 1), and we began systematically monitoring them approximately one year later. The *V. spectrum* bats occupy the highest section of the hollow tree, positioned approximately 4 to 4.5 meters above the ground, while a colony of 15 smaller bats, *Saccopteryx bilineata,* inhabits a lower section near the entrance of the tree hole, around 3 meters above ground level, maintaining spatial separation from *V. spectrum*. Although we did not individually mark the four *V. spectrum* bats we are confident that this colony consisted of a female and male bat pair and their respective two young for several reasons: 1) this species lives in a socially monogamous social structure that either constitutes a single roosting bat or family group [42], 2) the bats in the video displayed noticeable size differences and the teats of the adult, post-lactating female were visible on occasion, and 3) additional video recordings in 2024 showed that after the older pup/subadult left the colony a new pup was born soon after and nursed by its mother and that the other large individual was male (because the penis was visible). The newborn pup was identified as male from further video recordings in 2024. The group size never exceeded four individuals in the roost at the same time.

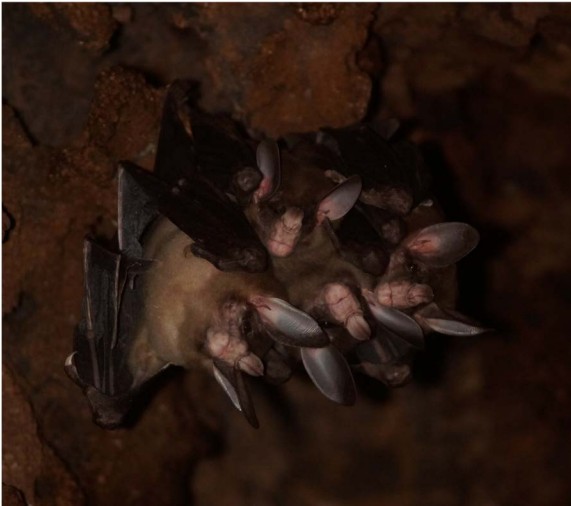

**Fig 1. Photo of the bats.** Roost with the four *Vampyrum spectrum* individuals (presumably male, female, and two pups).

## Data collection

At the beginning of the dry season in November 2023, a wildlife camera (Snapshot Mini Black 30MP 4K, Dörr GmbH) was installed inside the roost at a height of approximately 10 meters, using a small camera mount (CAMVATE photography mount with ¼"-20 thread). The camera was positioned about 2 meters from the primary roosting site of the bats, oriented upwards to capture optimal footage from below. To ensure minimal disturbance to the bats, the camera was carefully installed during a period of low activity. Observations following the installation indicated that the bats' behavior remained unaffected by the presence of the equipment. Over the course of three months, the camera was regularly checked and cleaned at least once a week, as well as equipped with a new SD card, and fitted with fresh batteries before being rein-stalled at the fixed location within the tree when necessary. The camera recorded automatically triggered infrared videos with a maximum duration of one minute. Initially, continuous recording was achieved each night; however, as battery power depleted, the camera's built-in power-saving function reduced the length of recorded videos.

Over the course of three months, we video-recorded the four bats for a total of 60 days. On some days, only a few videos were captured, while on other days, up to 40 videos were recorded, depending on how frequently the camera sensor was triggered by bat activity. In total, 502 videos were collected. Most of these videos (203 out of 502) were triggered by bats either leaving or entering the roost. The other bat species, *S. bilineata,* was very seldom recorded, as the camera was installed above their roosting spot. Out of the 502 recorded videos, 73 contained social behaviors or other interesting activity. By visual examina-tion of the videos, we classified eight different behavioral categories: "social roosting", "greeting behavior", "presumably sexual behavior", "bringing prey into the roost", "eating a prey item", "prey provision", "food checks", and "play behavior". All observed behaviors were recorded during the night between 5 pm and 6am and occurred with no specific distribution pattern (Fig 2). In order to ensure long-term monitoring of this social group, further videos are recorded in the colony in 2024/2025. All data pre-sented in this study are based on video recordings collected during the 2023/2024 season. However, one additional video from the 2024/2025 season was included as an anecdotal observation due to its particularly interesting and relevant nature.

## Individual distinction by size estimation

Due to the absence of individual markings on the animals, we employed supplementary methods for the distinction of individuals (adult vs. pup) to support our observations and interpretations of behaviors, particularly in prey-provision

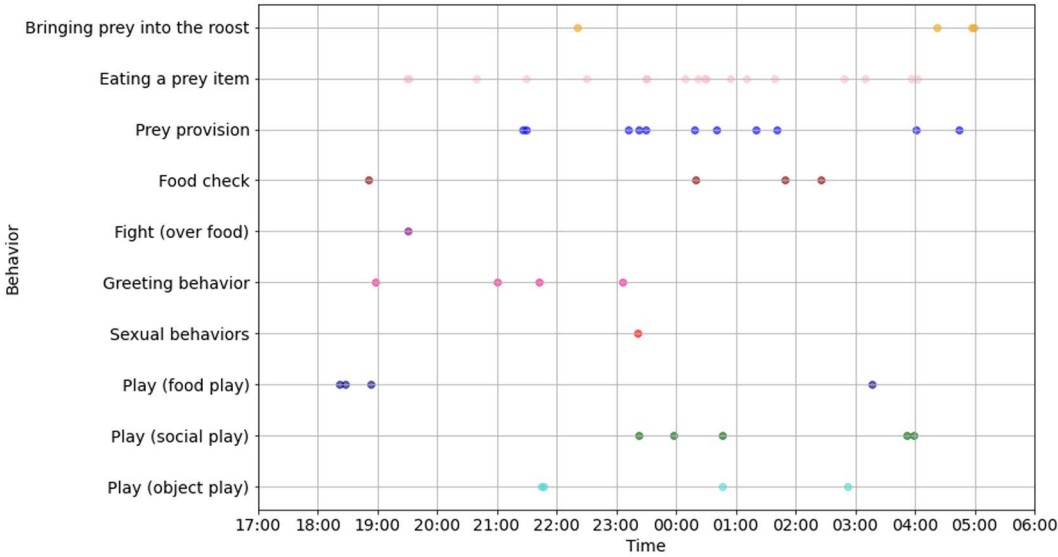

**Fig 2. Occurrence of social behaviors.** Scatter plot illustrating the temporal occurrence of the different observed social behaviors in a *V. spectrum* night-roost between 5 pm and 6 am (N = 60 nights). The behavior "social roosting" was excluded from this analysis because it occurred frequently both during the day and night, typically lasted for extended periods, and often lacked a clear endpoint, making it difficult to determine when the behavior concluded.

interactions. We measured the distance between the eyes of the bat entering the roost with prey and the bat approaching to take over the prey. This was done using screenshots from video frames capturing prey provision behaviors, where both animals were positioned at approximately the same height and their eyes were clearly visible (N = 8 prey- provision interactions). The screenshots were analyzed in GIMP (GIMP 2.10.38), utilizing the tape measure function to determine the eye distance in pixels. These measurements were subsequently compared using a non-parametric paired samples Wilcoxon test. Additionally, we used a spaghetti plot to visualize the results (Fig 3).

All statistical analyses were performed using the program Sypder (Spyder IDE, The Scientific Python Development Environment, Python 3.12.4).

## Results

Based on the 73 infrared videos, we created an ethogram that contained eight different behavioral categories exhibited by *V. spectrum* bats in their roost. Video examples of the respective behaviors can be found in the electronic supplementary material.

Social roosting (16 out of 73 videos): This behavior involves two or more bats roosting in close proximity with body contact, establishing a ball-like formation. At least one bat wraps its wings around the other bat(s). This behavior is often accompanied by allogrooming and/or social vocalizations and was observed very frequently.

Greeting Behavior (4 out of 73 videos): This behavior constitutes of a hugging-like interaction between a bat already in the roost and a newly arrived bat. The resident bat may actively approach or greet the newcomer as it reaches close proximity in the main roosting area. The greeting behavior is comparable to the initiation to social roosting, where at least one bat wraps its wings around the other, establishing a ball-like formation for several seconds (Fig 4). This behavior is often accompanied by social vocalizations.

Presumably sexual behavior (2 out of 73 videos): This behavior involves two bats, with one bat—presumably the male—positioned behind the female, aligning belly to back while wrapping its wings around her. This interaction often

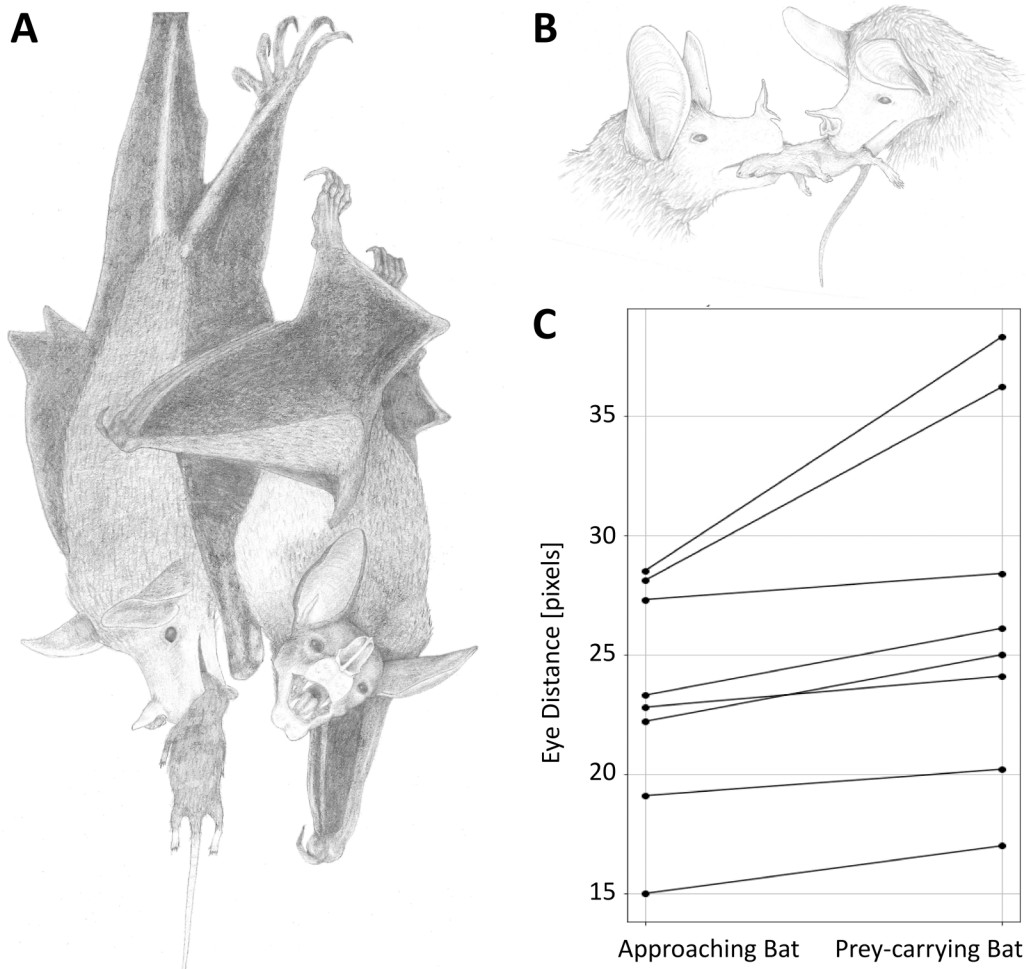

**Fig 3. Illustrations and eye distance measurements.** A+B: Illustrations of prey provision behavior between two bats (Illustration credit: Emma Dittrich); C: Spaghetti plot illustrating the difference of eye distances between the prey carrying bat and the approaching bat, showing a significant difference in eye distance (as a proxy for body size).

includes frequent position changes and licking of various lower body parts. This kind of behavior was only recorded twice and therefore we can only assume that this observed behavior is indeed produced in a sexual context.

Bringing prey into the roost (4 out of 73 videos): A bat enters the roost after foraging, carrying a prey item in its mouth (normally a bird or rodent). The prey is always held by its head and appears already dead. The bat carries the prey item in its mouth and crawls backward to the main roosting area in the hollow tree, with only the back or tail of the prey visible. The behavior classified here as "bringing prey into the roost" refers to video sequences in which a single bat was observed carrying prey into the cavity, but the subsequent interaction—whether the bat consumed the prey itself or shared it with another individual—could not be determined.

Eating a prey item (18 out of 73 videos): A bat suspends itself at a specific location within the roost, positioning the prey item between its wings and securing it with its thumb claws while biting into it. Audible chewing noises are a distinctive feature of this process. Prey items were identified as either small mammals (rats or mice) or birds. We estimate that consuming a single prey item may take approximately 30–40 minutes, based on sequential video recordings showing

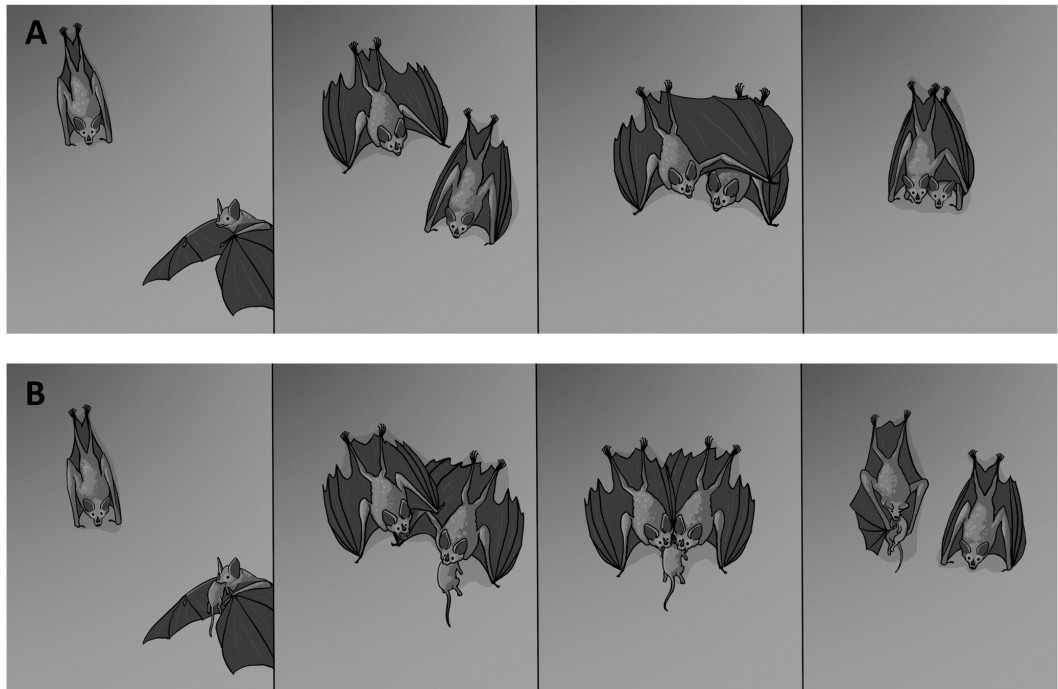

**Fig 4. Sequence of social behavior.** A: Illustration of greeting behavior; B: Illustration of prey-provision behavior (Illustration credit: Paulo C. Ditzel).

bats feeding on the same prey item. Nevertheless, this duration remains unverified due to the current limitation of our video equipment, which recorded only 1-minute segments. A video sequence was classified as "eating a prey item" when there was no immediate social interaction with another bat preceding or following the behavior. In 12 videos, we could not observe the bat entering the roost, only the eating process was recorded, while in six videos a bat entered the roost with prey, chose not to share it, and consumed it alone.

Prey provision (12 out of 73 videos): This behavior involves a bat entering the roost with a prey item held in its mouth. In many instances, as soon as the prey-carrying bat climbs to the main roosting spot, normally only one other bat approaches while emitting vocalizations. The approaching bat comes very close to the prey-carrying bat, typically from the back and/or side. The approaching bat attempts to bite into the upper body of the prey. In most cases (92%), the prey-carrying bat voluntarily releases the prey item quickly. Subsequently, both bats crawl up to the main roosting location where the bat with the prey usually begins to consume the prey item immediately (Figs 3 and 4). On a single occasion, a recorded interaction between two bats involved a fight over prey brought back into the roost, resembling a tug-of-war scenario. Additionally, we recorded an anecdotal observation in a video from December 2024, where a bat—presumed to be the mother—was present in the roost alongside a small pup when another bat, presumed to be the adult male based on size similarity, entered carrying prey. The mother approached the incoming bat, swiftly and silently took the prey, and then returned to the primary roosting spot next to the pup with the prey to consume it.

Food checks (4 out of 73 videos): One bat enters the roost, and another bat approaches, seemingly to check if the entering bat has brought back food from foraging. The approaching bat displays behaviors similar to prey-provision interactions, though in these cases, no prey is present. After realizing this, both bats typically crawl up to the main roosting spot.

Play (13 out of 73 videos): This bat species exhibits notable curiosity and playful behavioral traits. Various play behaviors were observed and categorized as object play (n=4), social play (n=5), or food play (n=4). One individual was

observed interacting with the wildlife camera, occasionally altering its position while emitting vocalizations which we categorized as object play. In other videos, bats were attempting to catch passing cockroaches, classified as food play. Additional footage displayed playful interactions between two bats, characterized by playful fighting, which we categorized as social play.

### Prey determination

In 30 videos, a bat carrying and/or eating a prey item was observed. In 50% of the instances, the prey was a small mammal, primarily mice and rats. In 30% of the cases, the prey was visually identified as a bird. In 20% of the cases, the prey item could not be identified due to low video quality and/or the bats being too far from the camera.

### Entering/leaving the roost

A significant portion of the videos (203 in total) captured by the camera showed the bats entering and leaving the roost, as this is when they passed the camera and triggered the sensor. Although it appears that bats usually foraged alone, on 40 occasions, at least two bats were observed leaving or returning to the roost together, sometimes even all four bats. In six instances, two bats were observed leaving the roost simultaneously and returning together after an average of about 40 minutes.

### Eye distance measurements

We detected a clear difference in eye distance between bats involved in prey-provision interactions. The bat that brought the prey into the roost had a significantly larger eye distance than the bat receiving the prey (paired Wilcoxon test, $Z = −2.527$; $n = 8$, exact p: 0.0078; Fig 3C). Assuming that eye distance can be used as a proxy for body size, our data suggest that the bat provision prey is larger than the bat receiving prey, which would be expected if adults share prey with their offspring.

## Discussion

Here, we provide the first comprehensive account of prey provision and other social behaviors in the spectral bat *V. spectrum*. Prey provision was a clearly cooperative social behavior wherein a bat successfully captured prey, brought it to the roost where group members were present, and willingly transferred the prey to another bat. Our video observations consistently showed that upon entering the roost with prey, the carrying bat was often approached by another bat attempting to acquire the prey. When other bats were present, they typically remained calm and inactive, suggesting they recognized the intended recipient of the prey, often a bat already positioned to receive it. In most instances of prey provision, the providing bat voluntarily and promptly handed over the prey to the receiving bat. Occasionally, conflicts over prey arose, as evidenced by one video depicting a tug-of-war scenario where the approaching bat successfully wrestled the prey away from the original possessor. Previous observations of a *V. spectrum* pair with a pup in captivity suggest that male bats may occasionally bring prey to the roost, potentially to provision females (Bradbury, personal observation via [43]). This behavior aligns with anecdotal observations from December 2024, where one bat, presumably the lactating female, remained in the roost with the pup while two other bats were out foraging. The prey transfer between the prey-carrying bat and the presumed mother, which was remarkably silent comparted to other prey transfers that were accompanied by social vocalizations supports the hypothesis of males provisioning females under certain circumstances.

To date, there has been scarce evidence of prey provision among bats. In the closely related sister species *Chrotopterus auritus,* individuals reportedly bring back prey into the roost and then share it with the other members (Rodrigo Medellín, personal communication, [46]. Two studies in captivity provide additional evidence for parental food provision in carnivorous species. In the fish-eating *Noctilio albiventris*, [47] observed that just prior to weaning at three months, captive mothers fed juveniles with masticated fish or mealworms from their cheek pouches. Similarly, in *Megaderma lyra*, mothers reportedly

transferred either entire or partly consumed frogs to their young. These food transfers ceased when the young bats reached approximately 74 days of age [48]. It appears that lactating females of *M. lyra* provision their offspring by supplementing milk with solid food, akin to the behavior hypothesized for *V. spectrum* [43]. The only well-documented instance of prey provision behavior observed in the wild among bats occurs in *Micronycteris microtis*. In this species, weaned pups are provisioned with large insects by their mothers for an additional five months. This extended provisioning likely helps pups learn to handle challenging prey and refine the acoustic skills essential for effective hunting. Although *M. microtis* primarily forages on insects, it has a remarkably broad diet that also includes lizards, making it the smallest known carnivorous bat to date [49,50]. Another prominent example of food sharing behavior in bats is the well-studied regurgitation of blood in vampire bats; however, here, food is shared between kin as well as non-kin group members that are in critical physical conditions [51–53], rather than on a regular basis and/or for learning purposes. Nectar-feeding bats also regurgitate food, but only for their own offspring, presumably to substitute their diet on their way to independence [54].

### Extended pup dependency and presumed biparental care

Observations of a captive *V. spectrum* pair with a pup suggest that males may occasionally bring prey to the roost to provision females or nourish young pups not yet fully capable of independent hunting (Bradbury, via [43]). During the nursing period, pups likely rely on milk supplemented with meat, which facilitates their transition to a carnivorous diet and helps them develop prey-handling skills [43]. Our data supports this strategy as a way to ensure adequate nutrition and skill development in young bats. The additional video recordings from 2024 documented the presence of a newborn pup and revealed that nursing females frequently remained in the roost during the night with the young pup for approximately two months. During this period, the adult male was regularly observed bringing prey items into the roost and provide either the older pup or the female with food. Although the absence of individual identification precludes definitive assessments of the relative contributions of each parent, the available evidence suggests that mainly males bring in the prey item and then share it with another individual.

Based on eye-distance measurements, two smaller individuals in our roost likely represent pups from different years, as twins have never been reported in *V. spectrum*. Pups are reported to be born at the end of the dry season [42,43] and the roost with these four bats was already discovered by us in November 2022, so the younger offspring must have been born at least 1.5 years ago, and the older one approximately 2.5 years ago. This indicates a prolonged period of parental care, which is relatively uncommon in bats. This extended investment is likely tied to the species' socially monogamous structure, a rarity among mammals [55,56] and to the challenges associated with carnivory. Parental investment, defined as post-birth behaviors that enhance offspring reproductive success [57], is influenced by the costs and benefits to parents [58]. In species with low mate competition and high paternity certainty, such as *V. spectrum*, males are more likely to invest in parental care. This aligns with the species' low population density, large territory requirements, and limited roost availability in dry forests [59]. Biparental care, hypothesized by Vehrencamp et al. [43], likely enhances offspring survival and success compared to uniparental care [60]. While biparental care is common in 90% of bird species [61], it is rare in mammals, where females alone provide care in 90% of species [58]. Notably, some of the social behaviors of the spectral bat seem to have quite some commonalties with the big-eared woolly bat (*Chrotopterus auritus)* that also exhibits a socially monogamous social group structure, shares a similar habitat and comparable diet [62,63]. Therefore, it can be surmised that similar factors act on *C. auritus* and it would be very likely that this species has also evolved some kind of biparental care [46].

### Prey items and foraging dynamics

In addition to prey provision behavior, several videos documented bats consuming prey within the roost. Due to the 1-minute recording limitation, it was often unclear whether the prey was brought into the roost by the bat itself or transferred prior to the recording. However, some videos showed bats entering with prey and consuming it alone without

transferring. In one instance, a bat carrying prey crawled to its roosting spot and ate it despite a nearby bat approaching and gently stretching its head toward the prey, possibly begging. The prey-carrying bat appeared undisturbed and continued eating. Vehrencamp et al. [43] proposed that bats might bring prey to the roost when caught late at night, close to sunrise, to avoid daylight exposure. However, our observations recorded such behavior at various times during the night, suggesting that when prey is captured near the roost, it provides a convenient, safe location for consumption. Visual classifications identified small rodents as the primary prey brought back to the roost, with only 9 of 30 videos showing birds being brought in. A study in 2022 based on fecal sample analysis from a *V. spectrum* roost in Nicaragua found birds to be the most common prey group, followed by bats and then small rodents [59]. Interestingly, no videos documented *V. spectrum* bringing smaller bat species back to the roost. Bats may be hunted midair, unlike diurnal birds, which are likely located in their nests through olfaction [42,43]. While previous studies reported *V. spectrum* foraging solitarily, with individuals leaving and returning to the roost separately [43], our data recorded 40 instances where two or more bats left or returned simultaneously, including 6 cases where two bats left and returned together after approximately 40 minutes. These observations suggest that while *V. spectrum* typically forages alone, exceptions occur, particularly when subadults may forage with adults to practice and develop hunting skills.

### Further social behaviors

Play behavior may contribute to shaping adult skill sets. Observed examples in our videos include chasing cockroaches, exploring wildlife cameras, and play-fighting, likely among subadults. Play is often viewed as a "non-serious" behavior, where actions are performed for their own sake rather than practical outcomes. This allows youngsters to explore environments and develop responses in a low-risk manner [64,65]. Play and exploration are documented in humans and non-human animals, including birds, frogs, and cephalopods, but are most prevalent in mammals, especially those with extended parental care [65,66]. Species with greater parental care often exhibit more play, which helps juveniles develop adaptive behaviors for current and future challenges [67]. Play may also influence gene expression and evolution indirectly [65]. Our data highlights the presence of play behavior in bats, emphasizing its potential developmental importance, despite limited research to date.

The pronounced social roosting behaviors of *V. spectrum*, including hugging-like actions—characterized by embracing, wrapping, or holding closely with limbs, wings, or body—and greeting interactions, were observed daily. Most bats are highly social, roosting in close contact and relying on strong bonds for survival [8]. Such bonds may offer cooperative benefits, such as information transfer within the group. Similar to allogrooming, a behavior that helps maintain social bonds in dynamic systems [31,68], these close-contact behaviors in *V. spectrum* likely reinforce social ties. While all bats exhibit autogrooming, allogrooming is less common [69–71]. In *V. spectrum*, hugging and greeting likely strengthen bonds within their small social groups, which is crucial for both parent-offspring and male-female relationships.

### Conclusion

In conclusion, our findings revealed that prey provision in *V. spectrum* is a common and mostly voluntary behavior, with bats bringing prey back to the roost and handing it over to other roost-mates, particularly those that may be less capable of hunting for themselves. This behavior supports the hypothesis that adults assist in transitioning young bats from milk to a carnivorous diet, ensuring adequate food intake and offering them the opportunity to practice how to handle large prey items. This study contributes to the broader understanding of bat sociality by highlighting the unique cooperative behaviors of *V. spectrum* and provides a foundation for further research into the ecological and evolutionary implications of these behaviors. The insights gained from this study not only enhance our knowledge of *V. spectrum* but also contribute to the general understanding of social evolution and cooperative behaviors in mammals.

## Supporting information

**S1 File. Example video from above-described key social behaviors.** Prey provision behavior (small mammal).
(MP4)

**S2 File. Example video from above-described key social behaviors.** Prey provision behavior (bird).
(MP4)

**S3 File. Example video from above-described key social behaviors.** Two bats fight about food.
(MP4)

**S4 File. Example video from above-described key social behaviors.** Social roosting behavior.
(MP4)

**S5 File. Example video from above-described key social behaviors.** Greeting behavior.
(MP4)

**S6 File. Example video from above-described key social behaviors.** Play (object) behavior.
(MP4)

**S7 File. Example video from above-described key social behaviors.** Play (food) behavior.
(MP4)

**S8 File. Example video from above-described key social behaviors.** Play (social) behavior.
(MP4)

**S1 Table. Data occurrence social behaviors.** Frequency of social behaviours of *V. spectrum* (only night) and their occurrence without "social roosting" behavior.
(XLSX)

**S2 Table. Data eye distance measurements.** Pixel measurements for calculating the eye distance.
(XLSX)

**S3 Table. Roost departure and return data.** Time stamps indicating when and how many bats left or returned to the roost based on video recordings. Potential simultaneous departures and subsequent returns are highlighted using a consistent color scheme.
(XLSX)

## Acknowledgments

We would like to express our thanks to all the workers from "La Estación Experimental Forestal Horizontes", especially Marlon Gonzalez. We are also grateful to Emma Dittrich and Paulo C. Ditzel for their drawings.

## Author contributions

**Conceptualization:** Marisa Tietge, Mirjam Knörnschild.

**Data curation:** Marisa Tietge.

**Formal analysis:** Marisa Tietge.

**Funding acquisition:** Mirjam Knörnschild.

**Investigation:** Marisa Tietge.

**Methodology:** Marisa Tietge.

**Project administration:** Marisa Tietge.

**Software:** Marisa Tietge.

**Supervision:** Mirjam Knörnschild.

**Validation:** Marisa Tietge, Mirjam Knörnschild.

**Visualization:** Marisa Tietge.

**Writing – original draft:** Marisa Tietge.

**Writing – review & editing:** Marisa Tietge, Eduardo Artavia Durán, Mirjam Knörnschild.

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
