## [Decision Letter · Decision Letter 0]

PONE-D-25-10917Cooperative Behaviors and Social Interactions in the Carnivorous Bat Vampyrum spectrumPLOS ONE

Dear Dr. Tietge,

Thank you for submitting your manuscript to PLOS ONE. After careful consideration, we feel that it has merit but does not fully meet PLOS ONE’s publication criteria as it currently stands. Therefore, we invite you to submit a revised version of the manuscript that addresses the points raised during the review process.

We look forward to receiving your revised manuscript.

Kind regards,

Nickson E. Otieno

Academic Editor

PLOS ONE

Journal Requirements:

“This work was supported by a grant from the Leibniz Foundation (P122/2020) to Mirjam Knörnschild.”

4. We note that your Data Availability Statement is currently as follows: All relevant data are within the manuscript and in Supporting Information files.

6. Please upload a copy of Fig 4, to which you refer in your text on page 13. If the figure is no longer to be included as part of the submission please remove all reference to it within the text.

Additional Editor Comments:

Kindly address all the remaining minor issues highlighted by the reviewers, and in the revised final version of the article, provide a short account of how you have done so, or alternatively address each of them pint by point

Reviewers' comments:

Reviewer's Responses to Questions

**Comments to the Author**

1. Is the manuscript technically sound, and do the data support the conclusions?

Reviewer #1: Yes

Reviewer #2: Yes

2. Has the statistical analysis been performed appropriately and rigorously? 

Reviewer #1: Yes

Reviewer #2: Yes

3. Have the authors made all data underlying the findings in their manuscript fully available?

Reviewer #1: Yes

Reviewer #2: Yes

4. Is the manuscript presented in an intelligible fashion and written in standard English?

Reviewer #1: Yes

Reviewer #2: Yes

5. Review Comments to the Author

Reviewer #1: The manuscript titled “Cooperative Behaviors and Social Interactions in the Carnivorous Bat Vampyrum spectrum” presents observations from a single group of the charismatic and rarely studied Vampyrum spectrum. This is an exceptional and rare opportunity, and I believe the authors have taken full advantage of it. The manuscript makes a valuable contribution to our understanding of this elusive species.

Using relatively simple and straightforward methods, the authors describe a range of behaviors among group members—some of which are quite fascinating and, as noted, poorly documented in bats. The manuscript is clearly written, well organized, and the authors are careful not to over-interpret their findings. In my opinion, the manuscript is in excellent shape and requires only very minor revisions.

Below, I offer a few small suggestions for improvement. However, these are not essential, and the authors should feel free to disregard them if they prefer:

Line 76: You say “living in” twice.

In the methods section “Individual distinction by size estimation” it wasn’t immediately clear why this analysis was being conducted. Only later, in the results, did it become apparent that it was used to differentiate younger and older individuals. I suggest clarifying the purpose of this analysis earlier in the methods section.

Line 216: Remove “may”.

Line 288: “our data suggest that…”

Line 303: This sentence could be made clearer—perhaps: “…the approaching bat successfully wrestled the prey away from the original possessor.”

Discussion (Lines 356–368): After reading about biparental care, I found myself wondering whether you have any observations or data regarding the relative investment in pup care by the male and female. I understand that without tagging individuals this may be difficult, but if there is any indication—such as who provided prey items—it could be valuable to include here.

Overall, this is a well-executed and insightful manuscript that offers a rare glimpse into the social behavior of Vampyrum spectrum, and I believe it will be a valuable addition to the literature on bat sociality. Great job!

Reviewer #2: Only a few details that need to be provided to better understand the context and the background please. It is very important that they discuss their dry forest and more detail about the roost and the bats that compose the group

6. PLOS authors have the option to publish the peer review history of their article (what does this mean? ). If published, this will include your full peer review and any attached files.

**Do you want your identity to be public for this peer review?** For information about this choice, including consent withdrawal, please see our Privacy Policy .

Reviewer #1: **Yes: ** Gloriana Chaverri

Reviewer #2: **Yes: ** Rodrigo Medellin

---

## [Author Response · Author response to Decision Letter 1]

3 Jul 2025

Comments Reviewer #1:

Original comment: Line 76: You say “living in” twice

Revised version: Deletion of one “living in” (line 59)

Original comment: In the methods section “Individual distinction by size estimation” it wasn’t immediately clear why this analysis was being conducted. Only later, in the results, did it become apparent that it was used to differentiate younger and older individuals. I suggest clarifying the purpose of this analysis earlier in the methods section

Revised version: Added clarification to the original sentence: “Due to the absence of individual markings on the animals, we employed supplementary methods for the distinction of individuals (adult vs. pup) to support our observations and interpretations of behaviors, particularly in prey-sharing interactions”. (line 176)

Original comment: Line 216: Remove “may”

Revised version: Removed “may” and replaced it with “is often”. (line 206)

Original comment: Line 288: “our data suggest that…”

Revised version: Deletion of one “s” at the end: „our data suggest“ (line 282)

Original comment: Line 303: This sentence could be made clearer—perhaps: “…the approaching bat successfully wrestled the prey away from the original possessor.”

Revised version: Adoption of the proposed wording: Occasionally, conflicts over prey arose, as evidenced by one video depicting a tug-of-war scenario where the approaching bat successfully wrestled the prey away from the original possessor. (line 296-298)

Original comment: Discussion (Lines 356–368): After reading about biparental care, I found myself wondering whether you have any observations or data regarding the relative investment in pup care by the male and female. I understand that without tagging individuals this may be difficult, but if there is any indication—such as who provided prey items—it could be valuable to include here

Revised version: Addition of two new sentences for more supporting arguments:

“The additional video recordings from 2024 that documented the presence of a newborn pup, revealed that nursing females frequently remained in the roost during the night with the young pup for approximately two months. During this period, the adult male was regularly observed bringing prey items into the roost and sharing them either with the older pup or with the female. Although the absence of individual identification precludes definitive assessments of the relative contributions of each parent, the available evidence suggests that mainly males bring in the prey item and then share it with another individual”. (line 335-342)

Comments Reviewer #2:

Original comment: Line 121 and following. It says “amidst typical dry tropical forest vegetation”. I doubt that there is such a thing as typical dry tropical forest. The dry tropical forest of Costa Rica is definitely different than the dry tropical forest of Honduras, Guatemala, and Mexico. In fact it is so different that the dry tropical forest in Costa Rica allows species that live only in wet tropical forest (such as V spectrum) to live there. Please add a brief description of the vegetation and I would love to see a brief discussion about why V spectrum lives in tropical rainforest always except in Costa Rica and a few areas in South America (I understand, though, that this is not a biogeography or an ecology study, so if you don´t wish to do that, just add the vegetation description I ask).

Revised version: Clarification that we referred to the typical dry forest in Guanacaste, Costa Rica:

“The focal roost is located within a partially hollow yet living tree of the species Manilkara chicle, amidst typical dry forest vegetation of Guanacaste, Costa Rica and adjacent to a dried riverbed” (line 104).

Also, addition of a short description of the dry forest in this area: “This area of tropical dry forest is characterized by a stratified vegetation structure comprising canopy, understory, shrub, and ground layers, each supporting distinct plant and animal communities. The canopy layer ranges from 20 to 30 meters in height and is composed of deciduous trees with broad crowns. The understory consists of trees reaching 10 to 20 meters, typically with light canopies and slender trunks. Beneath this, the shrub layer (2–5 m) is dominated by thorny, multi-stemmed plants. (line 105-110)

Author additional comment: The study of Martínez-Fonseca et al. 2022 with V. spectrum was also conducted with a roost/a group situated in tropical lowland dry forest in Nicaragua.

Original comment: Please also add more details about the tree itself and its surroundings. Height? DBH? Is the canopy of this tree connected to others? Is this the tallest tree in the vicinity? Describe the hollow; for example, can it be accessed from the ground? Or is the entrance to the cavity up higher in the trunk? Dimensions of the entrance, height and diameter of the cavity (likely tapered, from how wide to how wide?) Is it a single cavity with one entrance only? How high are the V spectrum? The Saccopteryx? Is there a separation between them? How far apart?

Revised version: More details about the tree and roost: The roost site is located within a mature tree reaching a height of approximately 20 meters, with a maximum diameter at breast height (DBH) of 1m. The tree forms part of a structurally interconnected canopy, although it is not the tallest individual in the immediate vicinity. The roost itself is situated within a hollow trunk, the entrance of which is spanned approximately 2 meters above ground level. The opening is roughly 80 cm wide at its base, progressively narrowing to approximately 50 cm with height, and lacks any secondary openings along the trunk.” (line 110-116)

The V. spectrum bats occupy the highest section of the hollow tree, positioned approximately 4 to 4.5 meters above the ground, while a colony of 15 smaller bats, Saccopteryx bilineata, inhabits a lower section near the entrance of the tree hole, around 3 meters above ground level, maintaining spatial separation from Vampyrum spectrum. (line 120-123)

Original comment: Lines 128 and following. So it was originally four individuals and with the newborn five? Why is there no discussion of the other two individuals (assuming there were four and then five)? Sex? Please provide more detail as to the individual members of the group

Revised version: Addition of words/sentences for clarifying that there were always max. 4 bats present in the roost at the same time:

“3) additional video recordings in 2024 showed that after the older pup/subadult left the colony a newly born pup was born soon after and being nursed by its mother and that the other large individual was male (because the penis was visible). The newborn pup was identified as male from further video recordings in 2024. The group size never exceeded four individuals in the roost at the same time.” (line 128-132)

Author additional comment: The confusion may have arisen from the mentioning of additional video recordings from 2024, despite the primary data presented in the paper originating exclusively from 2023. The 2024 material was referenced solely as anecdotal material supporting information and is not part of the analysed dataset.

Original comment: Line 145. The camera was regularly checked, how often?

Revised version: Addition of the time indication: “Over the course of three months, the camera was regularly checked and cleaned at least once a week, as well as equipped with a new SD card, and fitted with fresh batteries before being reinstalled at the fixed location within the tree when necessary.” (line 144-147)

Original comment: Line 157. I rest my case. The reader cannot know where are the two bat species

Revised version: No direct changes in the text/ see revised parts above

Original comment: Line 208. Only four prey were seen being carried in? That is odd for these bats. Did you collect any remains from under the bats? Or I guess it´s limitations of your video equipment.

Revised version: Additional sentence to clarify: The behavior classified here as “bringing prey into the roost” refers to video sequences in which a single bat was observed carrying prey into the cavity, but the subsequent interaction—whether the bat consumed the prey itself or shared it with another individual—could not be determined (line 217-220).

Author additional comment: This limitation is most likely due to the restricted duration of the recordings, which were limited to one-minute clips.

Prey provision behavior was recorded on 12 occasions; however, it is likely that this behavior occurred more frequently than documented, as it may have gone unrecorded due to several limitations of the camera setup.

During each visit to the roost, we collected all animal remains found on the ground, which primarily consisted of bird feathers and, occasionally, small mammal bones. These samples are currently undergoing analysis and will be presented in a future publication, with the aim of providing more detailed insights into the dietary composition of this group.

Original comment: Line 216. Says “may takes” should say “may take”

Revised version: Grammar correction: word “take” now without the “s”. (line 225)

Original comment: Lines 208 and following, can you tell us a bit about the prey species being eaten in all these videos? Itching!

Revised version: Additional short sentence regarding the prey species: “Prey items were identified as either small mammals (rats or mice) or birds.” (line 223/224)

Author additional comment: Species level identification of prey will hopefully achieve as soon as the collected regularly remains are fully analysed. The video quality forbid further identification other than “small mammal” or “bird”.

Original comment: 227. Emitting vocalizations. Did you see some physical movement of the wings? In my experience both Chrotopterus and Vampyrum “beg” by vocalizing and also by shaking their wings noticeably. Sadly your videos do not catch the start of the interaction. In our experience, even before the adult carrying the prey shows in the camera field of view, the recipient (it can also be an adult!!) begins shaking its wings and vocalizing. You did not witness or record this?

Revised version: No direct changes in the text

Author additional comment: Additional video recordings captured further instances of prey-sharing, including the initiation of the interaction. However, our footage does not reveal clear wing movements associated with prey transfer. Instead, we observed whole-body movements by the receiving bat as it approached the individual carrying the prey. This may be attributable to the spatial constraints of the roost, which is relatively confined, as well as the low position of the entrance. Incoming bats must crawl upward toward the main roosting area while holding prey in their mouths, potentially limiting the expression of more conspicuous behavioral gestures.

We agree that the recipient bat can also be an adult (as mentioned in the discussion part line 330- 338) and in at least two videos from 2024 it seems to be the case that one adult individual (male) is sharing with another adult individual (female), especially in times when the female is nursing a young pup like it was the case in nov/dec 2024.

Originally it was planned to catch the animals from this roost during the dry season in 2024 and non-inversive mark them but mainly due to bad weather conditions it was not possible for us. We will sure try again in the upcoming season end of 2025.

With a song meter bat mini all vocalisations from the bats during the night were separately recorded during these 3 months. We are currently analysing the vocalizations. We are hoping to put together the prey-sharing behavior with the respective vocalisations that are typically emitted in this context. This will be the topic of a further publication.

Original comment: 228 says “the bat sharing prey” it would be more accurate to say “the bat providing”. In the end, the provider and the recipient are sharing the prey

Revised version: We agree that “prey provision” is more accurate in this context because none of the recorded videos show actual joint consumption of the prey item. Rather, the behavior appears to involve a full transfer of the prey from one individual to another. In all observable cases (except the tug-of war-video), the prey-carrying bat hands over the entire prey item without retaining any portion for itself. This interpretation is supported by the visible chewing behavior: the receiving bat begins chewing the prey, while the transferring individual does not engage in feeding and typically proceeds to rejoin the others at the main roosting site.

---

## [Editor Report · Decision Letter 1]

Cooperative Behaviors and Social Interactions in the Carnivorous Bat Vampyrum spectrum

PONE-D-25-10917R1

Dear Dr. Tietge,

We’re pleased to inform you that your manuscript has been judged scientifically suitable for publication and will be formally accepted for publication once it meets all outstanding technical requirements.

Kind regards,

Nickson Erick Otieno, PhD

Academic Editor

PLOS ONE

Additional Editor Comments (optional):

The two reviewers have submitted their reports on your article and they both are of the opinion that after the revision by the authors, the manuscript now meets not only the standards of rigorous bat ecology research but also makes an important and timely contribution to understanding the nexus between bat feeding ecology  and their social dynamics. They also feel and the editors agree that the technical quality of the manuscript is now also at a level acceptable to PLOS ONE, if the authors are willing to make the remaining few minor changes. These suggested changes are highlighted within the reviewer’s reports accessible to the authors through the journal’s submission platform

In addition:

The title: Please ensure only the first word has a capitalized first letter, unless it is a proper noun, major place-name or genus in scientific name

Line 298: Please consider replacing ‘wrestled’ with ‘wrested’

Line 326-327: Please consider replacing “…substitute their diet on their way to independence” with “…for purposes of weaning”

In the Conclusion section, could the authors use a few lines to venture into potential future research on the topic, perhaps what they were unable to achieve within the constraints of the project’s resources?

Line 635: Please either ‘behaviour’ or ‘behavior’ consistently throughout (preferably the latter)

Please add the list of Figure legends in the manuscript body just before the Supporting information section
---

## [Editor Report · Acceptance letter]

PONE-D-25-10917R1

PLOS ONE

Dear Dr. Tietge,

I'm pleased to inform you that your manuscript has been deemed suitable for publication in PLOS ONE. Congratulations! Your manuscript is now being handed over to our production team.

Kind regards,

on behalf of

Dr. Nickson Erick Otieno

Academic Editor

PLOS ONE